# Graph-Time Convolutional Autoencoders

**Mohammad Sabbaqi**   **Riccardo Taormina**   **Alan Hanjalic**   **Elvin Isufi**

Delft University of Technology
{m.sabbaqi, r.taormina, a.hanjalic, e.isufi-1}@tudelft.nl

## Abstract

We introduce graph-time convolutional autoencoder (GTConvAE), a novel spatiotemporal architecture tailored to unsupervised learning for multivariate time series on networks. The GTConvAE leverages product graphs to represent the time series and a principled *joint* spatiotemporal convolution over this product graph. Instead of fixing the product graph at the outset, we make it parametric to attend to the spatiotemporal coupling for the task at hand. On top of this, we propose temporal downsampling for the encoder to improve the spatiotemporal receptive field without affecting the network structure; respectively, in the decoder, we consider the opposite upsampling operator. We prove that the GTConvAEs with graph integral Lipschitz filters are stable to relative network perturbations, ultimately showing the role of the different components in the encoder and decoder. Numerical experiments for denoising and anomaly detection in solar and water networks corroborate our findings and showcase the effectiveness of the GTConvAE compared with state-of-the-art alternatives.

## 1 Introduction

Learning unsupervised representations from spatiotemporal network data is commonly encountered in multivariate data denoising [1], anomaly detection [2], missing data imputation [3], and forecasting [4, 5], to name just a few application areas. The challenge is to develop models that *jointly* capture the spatiotemporal dependencies in a computation- and data-efficient manner yet being tractable so that to understand the role played by the network structure and the dynamics over it. The autoencoder family of functions is of interest in this setting, but vanilla spatiotemporal forms [6–8] that ignore the network structure suffer the well-known curse of dimensionality and lack inductive learning capabilities [9].

Upon leveraging the network as an inductive bias [10], graph-time autoencoders have been recently developed. These approaches are typically composed of two interleaving modules: one capturing the spatial dependencies via graph neural networks (GNNs) [11] and one capturing the temporal dependencies via temporal CNN or LSTM networks. For example, the work in [1] uses an edge-varying GNN [12] followed by a temporal convolution for motion denoising. The work in [13] considers LSTMs and graph convolutions for variational spatiotemporal autoencoders, which have been further investigated in [3, 14], respectively, for spatiotemporal data imputation as a graph-based matrix completion problem and dynamic topologies. Graph-time autoencoders over dynamic topologies have also been investigated in [15, 16]. Lastly, [4] embeds the temporal information into the edges of a graph and develops an autoencoder over this graph for forecasting purposes.

By working disjointly first on the graph and then on the temporal dimension of the graph embeddings, these approaches fail to capture the joint spatiotemporal dependencies present in the raw data. It is also challenging to analyze their theoretical properties and to attribute to what extent the benefit comes from one module over the other. These aspects have been investigated for supervised spatiotemporal learning via GNNs [17–23] but not for autoencoders. The two works elaborating on spatiotemporal autoencoders are [2] and [24]. The work in [2] replicates the graph over time via the Cartesian product principle [25] and uses an order one graph convolution [26] to learn spatiotemporal embeddings that

M. Sabbaqi et al., Graph-Time Convolutional Autoencoders. *Proceedings of the First Learning on Graphs Conference (LoG 2022)*, PMLR 198, Virtual Event, December 9–12, 2022.

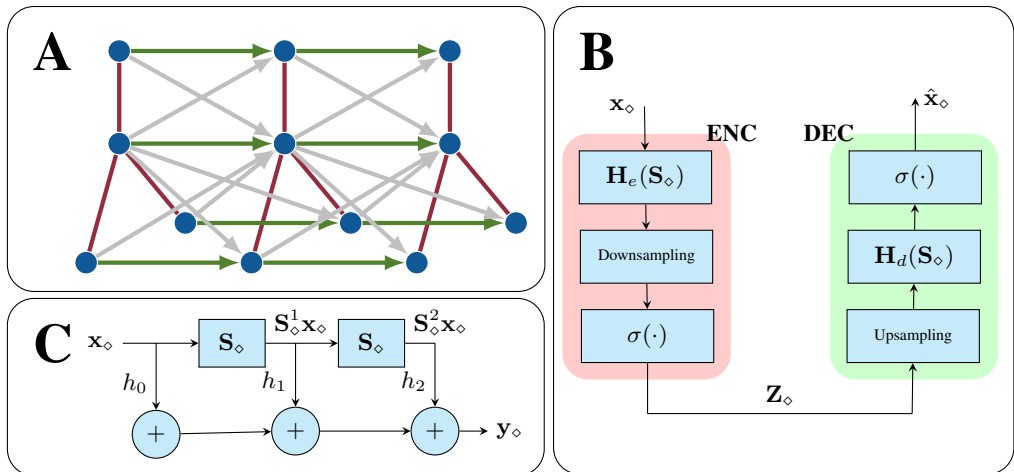

**Figure 1:** Graph-time convolution autoencoder components. (A) Parametric product graph representation of time series [cf. (1)]: red edges represent spatial connections and are related to $s_{01}$, green and gray edges represent temporal connections and are ruled by $s_{10}$ and $s_{11}$, respectively. (B) A single layer GTConvAE. The encoder and decoder are mirrored versions of each other containing: i) a graph convolutional filter $\mathbf{H}(\mathbf{S}_\diamond)$ [cf. (2)] operating w.r.t. the product graph shift operator $\mathbf{S}_\diamond$ for the time series $\mathbf{x}_\diamond$; ii) a temporal downsampling [cf. (4)] respectively upsampling [cf. (8)] module; and iii) a pointwise nonlinearity $\sigma(\cdot)$. (C) A second order graph-time convolutional filter. The shifted variants of input $\mathbf{x}_\diamond$ are linearly combined with weights $h_0, h_1, h_2$ to generate the filtred output $\mathbf{y}_\diamond$.

are fed into an LSTM module to improve the temporal memory, ultimately giving more importance to the temporal dimension of the latent representation. Differently, [27] proposed a variational graph-time autoencoder that its encoder is based on [18] and its decoder is a multi-layer perceptron; hence, being suitable only for topological tasks such as dynamic link prediction but not for tasks concerning time series over networks such as denoising or anomaly detection.

In this paper, we propose a GTConvAE that, differently from [2], captures *jointly* the spatiotemporal coupling both in the raw data and the intermediate higher-level representations. The GTConvAE operates over a parametric product graph [28] to attend to the spatiotemporal coupling for the task at hand rather than fixing it at the outset. Differently from [18], the GTConvAE has a symmetric structure with graph-time convolutions in both encoder and decoder, making it suitable for tasks concerning network time series. We also study the capability of the GTConvAE to transfer learning across different networks, which is of importance as practical topologies differ from the models used during training (e.g., because of model uncertainness, perturbations, or dynamics). The latter has been studied for traditional [29–31] and graph-time GNN models [21, 28, 32] but not for graph-time autoencoders.

Our contribution in this paper is twofold. First, we propose a symmetric graph-time convolutional autoencoder that jointly captures the spatiotemporal coupling in the data suited for tasks concerning multivariate time series over networks. The GTConvAE represents the time series as a graph signal over product graphs and uses the latter as an inductive bias to learn unsupervised representations. The product graph is parametric to attend to the coupling for the specific task, and it generalizes the popular choices of product graphs [33]. We also propose a temporal downsampling/upsampling in the encoder/decoder to increase the spatiotemporal receptive field without affecting the network structure; hence, preserving the inductive bias. Second, we prove GTConvAE is stable to relative perturbations on the spatial graph; highlighting the role played by the encoder, decoder, parametric product graph, convolutional filters, and downsampling/upsampling rate. Numerical experiments about denoising and anomaly detection over solar and water networks corroborate our findings and show a competitive performance compared with the more involved state-of-the-art alternatives.

The rest of this paper is organized as follows. Section 2 formulates the GTConvAE model and Section 3 analyzes its theoretical properties. Numerical experiments are presented in Section 4 and conclusions in Section 5. The proofs are collected in the appendix.

## 2   Graph-Time Convolutional Autoencoders

GTconvAE learns representations from $N-$dimensional multivariate time series $\mathbf{x}_t \in \mathbb{R}^N, t = 1, \ldots, T$, collected in matrix $\mathbf{X} \in \mathbb{R}^{N \times T}$. These time series have a spatial network structure represented by a graph $\mathcal{G} = (\mathcal{V}, \mathcal{E})$ composed of $N$ nodes $\mathcal{V} = \{v_1, \ldots, v_N\}$ and $M$ edges. The $n$-th row of $\mathbf{X}$ contains the time series $\mathbf{x}^n = [x_1(n), \ldots, x_T(n)]^\top$ on node $v_n$ and the $t$-th column a graph signal $\mathbf{x}_t = [x_t(1), \ldots, x_t(N)]^\top$ at timestamp $t$ [34, 35]. For example, the time series could be nodal pressures measured over junction nodes in a water distribution network, while the pipe connections rule the spatial structure. The representations learned from the tuple $\{\mathcal{G}, \mathbf{X}\}$ can then be used, among others, for anomaly detection [6], denoising dynamic data over graphs [1], and missing data completion [3].

The GTconvAE follows the standard encoder-decoder structure [36], but in each module, it *jointly* captures the spatiotemporal dependencies in the data. We denote the GTconvAE as

$$\hat{\mathbf{X}} = \text{GTConvAE}(\mathbf{X}, \mathcal{G}; \mathcal{H}) := \text{DEC}\big(\mathbf{Z} := \text{ENC}(\mathbf{X}, \mathcal{G}; \mathcal{H}_e), \mathcal{G}; \mathcal{H}_d\big)$$

where the encoder $\text{ENC}(\cdot, \cdot; \mathcal{H}_e)$ and decoder $\text{DEC}(\cdot, \cdot; \mathcal{H}_d)$ are non-linear parametric functions and where set $\mathcal{H} = \mathcal{H}_e \cup \mathcal{H}_d$ collects all parameters. The encoder takes as input the graph $\mathcal{G}$ and the time series $\mathbf{X}$ and produces higher-level representations $\mathbf{Z} \in \mathbb{R}^{N \times T_e}$. These representations are built in a layered manner where each layer comprises: $i$) a *joint* graph-time convolutional filter to capture the spatiotemporal dependencies in a principled manner; ii) a temporal downsampling module to increase the receptive field without affecting the network structure; and iii) a pointwise nonlinearity to have more complex representations. The decoder has a mirrored structure w.r.t. the encoder by taking as input $\mathbf{Z}$ and outputting an estimate of the input $\hat{\mathbf{X}}$. The model parameters are estimated end-to-end by minimizing a spatiotemporal regularized reconstruction loss $\mathcal{L}(\mathbf{X}, \hat{\mathbf{X}}, \mathcal{G}, \mathcal{H})$.

### 2.1   Product Graph Representation of Network Time Series

GTConvAE uses product graphs to represent the spatiotemporal dependencies in $\mathbf{X}$ [25]. Product graphs have been proven successful for processing multivariate time series, such as imputing missing values [37, 38], denoising [39], providing a spatiotemporal Fourier analysis [35], as well as building vector autoregressive models [40], spatiotemporal scattering transforms [41], and graph-time neural networks [28]. Specifically, denote by $\mathbf{S} \in \mathbb{R}^{N \times N}$ the graph shift operator (GSO) of the spatial graph $\mathcal{G}$, e.g., adjacency, Laplacian. Consider also a temporal graph $\mathcal{G}_T = (\mathcal{V}_T, \mathcal{E}_T, \mathbf{S}_T)$, where the node set $\mathcal{V}_T = \{1, \ldots, T\}$ comprises the discrete-time instants, the edge set $\mathcal{E}_T \subseteq \mathcal{V}_T \times V_T$ captures the temporal dependencies; e.g., a directed line or a cyclic graph, and $\mathbf{S}_T \in \mathbb{R}^{N \times N}$ is the respective GSO [42, 43]. The time series $\mathbf{x}^n$ now can be defined as a graph signal over temporal graph $\mathbf{S}_T$ where $x_t(n)$ is a scalar value assigned to the $t$-th node of $\mathcal{G}_T$.

The product graph representing the spatiotemporal patterns in $\mathbf{X}$ is denoted by $\mathcal{G}_\diamond = \mathcal{G}_T \diamond \mathcal{G} = (\mathcal{V}_\diamond, \mathcal{E}_\diamond, \mathbf{S}_\diamond)$. The node set $\mathcal{V}_\diamond$ is the Cartesian product between $\mathcal{V}_T$ and $\mathcal{V}$ which leads to $NT$ distinct spatiotemporal nodes $i_\diamond = (n, t)$. The edge set $\mathcal{E}_\diamond$ connects these nodes and the GSO $\mathbf{S}_\diamond \in \mathbb{R}^{NT \times NT}$ is dictated by the product graph. Fixing the product graph implies fixing the spatiotemporal dependencies in the data, which may lead to wrong inductive biases. To avoid this and improve flexibility, we consider a parametric product graph whose GSO is of the form

$$\mathbf{S}_\diamond = \sum_{i=0}^{1} \sum_{j=0}^{1} s_{ij}(\mathbf{S}_T^i \otimes \mathbf{S}^j) = \underbrace{s_{00}\mathbf{I}_T \otimes \mathbf{I}_N}_{\text{self-loops}} + \underbrace{s_{01}\mathbf{I}_T \otimes \mathbf{S} + s_{10}\mathbf{S}_T \otimes \mathbf{I}_N}_{\text{Cartesian}} + \underbrace{s_{11}\mathbf{S}_T \otimes \mathbf{S}}_{\text{Kronecker}}, \quad (1)$$

$$\underbrace{\phantom{s_{01}\mathbf{I}_T \otimes \mathbf{S} + s_{10}\mathbf{S}_T \otimes \mathbf{I}_N + s_{11}\mathbf{S}_T \otimes \mathbf{S}}}_{\text{Strong}}$$

where the scalar parameters $\{s_{ij}\}$ attend the spatiotemporal connections and encompass the typical product graph choices such as the Kronecker, the Cartesian, and the strong product. By column-vectorizing $\mathbf{X}$ into $\mathbf{x}_\diamond = \text{vec}(\mathbf{X}) \in \mathbb{R}^{NT}$, we obtain a product graph signal assigning a real value to each spacetime node $i_\diamond$. i.e., the dynamic data $\mathbf{x}_t$ over $\mathcal{G}$ is now a static signal $\mathbf{x}_\diamond$ over the product graph $\mathcal{G}_\diamond$.

### 2.2   Encoder

The encoder is an $L_e$-layered architecture in which each layer comprises a bank of product graph convolutional filters, temporal downsampling, and pointwise nonlinearities.

**GTConv filter** captures the spatiotemporal patterns in the data matrix $\mathbf{X}$. Given the parametric product graph representation $\mathcal{G}_\diamond = (\mathcal{V}_\diamond, \mathcal{E}_\diamond, \mathbf{S}_\diamond)$ [cf. (1)] and the product graph signal $\mathbf{x}_\diamond = \text{vec}(\mathbf{X})$ as input, the output of a graph-time convolutional filter of order $K$ is

$$\mathbf{y}_\diamond = \mathbf{H}(\mathbf{S}_\diamond)\mathbf{x}_\diamond = \sum_{k=0}^{K} h_k \mathbf{S}_\diamond^k \mathbf{x}_\diamond \tag{2}$$

where $\mathbf{h} = [h_0, \ldots, h_K]^\top$ are the filter parameters and $\mathbf{H}(\mathbf{S}_\diamond) := \sum_{k=0}^{K} h_k \mathbf{S}_\diamond^k$ is the filtering matrix. The filter in (2) is called convolutional as the output $\mathbf{y}_\diamond$ is a weighted linear combination of shifted graph signals over the product graph up to $K$ times [44]. Hence, the filter is spatiotemporally local in a neighborhood of radius $K$. The filter locality does not only depend on the order $K$ but also on the type of product graph. For example, for a fixed $K$, the Cartesian product is more localized than the strong product, which can be considered to have a longer spatiotemporal memory [28]. Consequently, learning parameters $\{s_{ij}\}$ in (1) implies learning the multi-hop resolution radius.

In the $\ell$−th layer, the encoder has $F_{\ell-1}$ product graph signal features $\mathbf{x}_{\diamond,\ell-1}^1, \ldots, \mathbf{x}_{\diamond,\ell-1}^g, \ldots \mathbf{x}_{\diamond,\ell-1}^{F_{\ell-1}}$, processes these with a bank of $F_\ell F_{\ell-1}$ filters and outputs $F_\ell$ product graph signal features as

$$\mathbf{y}_{\diamond,\ell}^f = \sum_{g=1}^{F_{\ell-1}} \mathbf{H}^{fg}(\mathbf{S}_\diamond)\mathbf{x}_{\diamond,\ell-1}^g, \quad f = 1, \ldots F_\ell, \tag{3}$$

which are the higher-level linear representation of the layer.

**Temporal downsampling** reduces the temporal dimension in each output $\{\mathbf{y}_{\diamond,\ell}^f\}_f$ in (3) by downsampling the latter along the temporal dimension with a rate $r$. More specifically, we first transform the $f$−th output $\mathbf{y}_{\diamond,\ell}^f \in \mathbb{R}^{NT_{\ell-1}^e}$ into a matrix $\mathbf{Y}_1^f = \text{vec}^{-1}(\mathbf{y}_{\diamond,1}^f) \in \mathbb{R}^{N \times T_{\ell-1}^e}$ and then summarize every $r$ consecutive columns without overlap to obtain the downsampled matrix $\mathbf{X}_{d,\ell}^f \in \mathbb{R}^{NT_\ell^e}$ with $T_\ell^e < T_{\ell-1}^e$. The $(n,t)$−th entry of $\mathbf{X}_{d,\ell}^f$ is computed as

$$\mathbf{X}_{d,\ell}^f(n,t) = \text{SUM}\big(\mathbf{Y}_\ell^f(n, r(t-1)+1 : rt)\big), \quad f = 1, \ldots F_\ell, \tag{4}$$

where SUM$(\cdot)$ is a summary function over the temporal indices $r(t-1)+1$ to $rt$. This summary function could be a simple downsampling (i.e., output the first column in the block $\mathbf{Y}_\ell^f(n, r(t-1)+1 : rt)$) or an aggregation function (i.e., mean/max/min per spatial node).

This temporal downsampling increases the encoder spatiotemporal memory without affecting the spatial structure. I.e., nodes with the temporal indices $t, rt, (r+1)t, \ldots$ become neighbors, which brings in a longer memory in the next layer and increases the encoder receptive field. Temporal skip connections can be an alternative to increase the receptive field, however, the downsampling reduces the computational memory by shrinking the product graph and allows for a theoretical analysis. While also spatial graph pooling can be added [45], we do not advocate it for two reasons. First, the spatial graph acts as an inductive bias for the GTConvAE [10]; hence, changing the graph in the layers via graph reduction, coarsening, or alternatives will affect the spatial structure, ultimately changing the inductive bias. Second, the spatial graph often represents the communication channels for distributed implementation of GTConv [21, 46, 47], and changing it may be physically impossible as sensor nodes have a limited transmission radius. An option in the latter setting may be a zero-pad spatial pooling [48, 49] but it requires memorizing the indices where the zero-padding is applied, which may be challenging for large graphs.

**Activation functions** nonlinearize the downsampled features to increase the representational capacity. We consider an entry-wise nonlinear function $\sigma(\cdot)$ such as ReLU and produce layer $\ell$−th output as

$$\mathbf{X}_{\ell+1}^f = \sigma(\mathbf{X}_{d,\ell}^f), \quad f = 1, \ldots F_\ell. \tag{5}$$

The encoder performs operations (3)-(4)-(5) for all the $L_e$ layers to yield the encoded output

$$\mathbf{Z}_\diamond := \mathbf{X}_{\diamond,L} = \text{ENC}(\mathbf{x}_{\diamond,0}, \mathbf{S}, \mathbf{S}_T; \mathcal{H}_e, \mathbf{s}), \tag{6}$$

where $\mathbf{x}_{\diamond,0} := \mathbf{x}_\diamond \in \mathbb{R}^{NT}$, $\mathbf{Z}_\diamond = [\mathbf{z}_\diamond^1, \ldots, \mathbf{z}_\diamond^{F_L}] \in \mathbb{R}^{NT_{L_e} \times F_L}$, and we made explicit the dependence from the product graph parameters $\mathbf{s} = [s_{00}, s_{01}, s_{10}, s_{11}]^\top$ [cf. (1)].

## 2.3 Decoder

Mirroring the encoder, the decoder reconstructs the input from the latent representations in (6). At the generic layer $\ell$, graph-time convolutional filtering is performed, subsequently a temporal upsampling, and a pointwise nonlinearity.

**GTConv filtering** decodes the spatiotemporal latent representations from the encoder. Considering again $F_{\ell-1}$ input features $\mathbf{z}_{\diamond,\ell-1}^1, \ldots, \mathbf{z}_{\diamond,\ell-1}^g, \ldots, \mathbf{z}_{\diamond,\ell-1}^{F_\ell-1}$ and a filter bank of $F_\ell F_{\ell-1}$ GTConv filters as per (2), the outputs are

$$\mathbf{y}_{\diamond,\ell}^f = \sum_{g=1}^{F_{\ell-1}} \mathbf{H}^{fg}(\mathbf{S}_\diamond)\mathbf{z}_{\diamond,\ell-1}^g, \quad f = 1, \ldots F_\ell. \tag{7}$$

**Upsampling** zero-pads the removed temporal values during downsampling [cf. (4)] so that the final GTConvAE output matches the dimension of $\mathbf{X}$. Specifically, given the $f-$th feature $\mathbf{y}_{\diamond,\ell}^f \in \mathbb{R}^{NT_{\ell-1}^d}$ from (7), we again transform it into a matrix $\mathbf{Y}_1^f = \text{vec}^{-1}(\mathbf{y}_{\diamond,1}^f) \in \mathbb{R}^{N \times T_{\ell-1}^d}$ and obtain the upsampled matrix $\mathbf{Z}_{u,\ell}^f \in \mathbb{R}^{N \times T_\ell^d}$ whose $(n,t)-$th entry is computed as

$$\mathbf{Z}_{u,\ell}^f(n,t) = \begin{cases} \mathbf{Y}_\ell^f(n, \lceil t/r \rceil); & \text{if} \quad \exists k \in \mathbb{Z} : t = kr \\ 0; & \text{o/w} \end{cases} \tag{8}$$

where $\lceil \cdot \rceil$ is the ceiling function.[1] The GTConv filter bank in the next layer interpolates these zero-padded values from the downsampled ones. This implies that the downsampling rate in the encoder cannot be too harsh to lose information, and also, the filter orders in the decoder cannot be too small to have a high interpolatory capacity.

**Activation functions** again nonlinzearize the upsampled features in (8) and yield

$$\mathbf{Z}_\ell^f = \sigma(\mathbf{Z}_{u,\ell}^f), \quad f = 1, \ldots F_\ell. \tag{9}$$

The decoder performs operations (7)-(8)-(9) for all $L_d$ layers to yield the decoded output $\hat{\mathbf{x}}_\diamond = \mathbf{z}_{\diamond,L_d} \in \mathbb{R}^{NT}$, which also corresponds to the GTConvAE output

$$\hat{\mathbf{x}}_\diamond = \mathbf{z}_{\diamond,L_d} = \text{DEC}(\mathbf{Z}_\diamond, \mathbf{S}, \mathbf{S}_T; \mathcal{H}_d, \mathbf{s}), \tag{10}$$

where we match the dimensions by setting $F_{L_d} = 1$.

## 2.4 Loss Function

Given (6) and (10), the GTConvAE in (1) can be detailed as

$$\hat{\mathbf{x}}_\diamond = \text{GTConvAE}(\mathbf{x}_\diamond, \mathbf{S}, \mathbf{S}_T; \mathcal{H}, \mathbf{s}) = \text{DEC}\big(\text{ENC}(\mathbf{x}_\diamond, \mathbf{S}, \mathbf{S}_T; \mathcal{H}_e, \mathbf{s}), \mathbf{S}, \mathbf{S}_T; \mathcal{H}_d, \mathbf{s}\big). \tag{11}$$

The GTConv filter parameters in $\mathcal{H}$ and the product graph parameters in $\mathbf{s}$ are estimated by minimizing the loss function

$$\mathcal{L}(\mathbf{X}, \hat{\mathbf{X}}, \mathcal{G}, \mathcal{H}) = \mathbb{E}_\mathcal{D}\left[\|\mathbf{x}_\diamond - \hat{\mathbf{x}}_\diamond\|_2\right] + \rho\|\mathbf{s}\|_1. \tag{12}$$

where the first term measures the reconstruction error over the probabilistic distribution $\mathcal{D}$ of the training set, whereas the second term imposes sparsity in the spatiotemporal dependencies of the product graph. Scalar $\rho > 0$ controls the trade-off between fitting and regularization, and a higher value implies a stronger spatiotemporal sparsity (from the norm one $\|\cdot\|_1$); i.e., sparser spatiotemporal attention.

*Complexity analysis:* Denoting the maximum number of features in all layers by $F_{max} = \max\{F_\ell\}$ the GTConvAE has $|\mathcal{H}| = (L_e + L_d)(K+1)F_{max}^2$ parameters. This is because each GTConv filter (2) has $K+1$ parameters and in each layer a filter bank of at most $F_{max}^2$ filters is used. Despite the product graphs are of large dimensions, the latter is highly sparse and the computation complexity of the GTConvAE is of order $\mathcal{O}(M_\diamond|\mathcal{H}|)$, where $M_\diamond = NT + NM_T + MT + 2MM_T$ is the number of edges of the product graph ($M$ edges in the spatial graph and $M_T$ edges in the temporal graph). This is because each graph-time filter has a computational complexity of order $\mathcal{O}((K+1)M_\diamond)$ [28] and the GTConvAE consists of $(L_e + L_d)F_{max}^2$ graph-time filters. We consider $r = 1$ sampling rate to provide the worst case analysis, but the computational complexity reduces further for $r > 1$.

---

[1]We considered the same down/up-sampling rate in each layer of the decoder and encoder; hence, because of the mirrored structure $T_\ell^e$ in (5) equals $T_{\ell-1}^d$ in (8).

# 3 Stability Analysis

In this section, we conduct a stability analysis of the GTConvAE w.r.t. relative perturbations in the spatial graph. This stability analysis is motivated by the fact that we do not always have access to the ground truth spatial graph due to modeling issues or when the physical network undergoes slight changes over time. Hence, the spatial graph used for training differs from that used for testing; thus, having a stable GTConvAE is desirable to perform the tasks reliably.

We consider the *relative perturbation* model

$$\hat{\mathbf{S}} = \mathbf{S} + (\mathbf{S}\mathbf{E} + \mathbf{E}\mathbf{S}) \tag{13}$$

where $\hat{\mathbf{S}}$ is the perturbed GSO and $\mathbf{E}$ is the perturbation matrix with bounded *operator norm* $\|\mathbf{E}\| \leq \epsilon$ [29]. This model accounts for graph perturbation depending on its structure, i.e., a higher degree node (a node with higher-weighted connected edges) is relatively prone to more perturbation.

## 3.1 Spatiotemporal integral Lipschitz filters

To investigate the stability of GTConvAE, we first characterize the graph-time convolutional filters in the spectral domain. Consider the eigendecompositions of the spatial GSO $\mathbf{S} = \mathbf{V}\boldsymbol{\Lambda}\mathbf{V}^{\mathsf{H}}$ and of the temporal GSO $\mathbf{S}_T = \mathbf{V}_T\boldsymbol{\Lambda}_T\mathbf{V}_T^{\mathsf{H}}$. Matrices $\mathbf{V} = [\mathbf{v}_1, \ldots, \mathbf{v}_N]^{\top}$ and $\mathbf{V}_T = [\mathbf{v}_{T,1}, \ldots, \mathbf{v}_{T,T}]^{\top}$ collect the spatial and the temporal eigenvectors, respectively, and $\boldsymbol{\Lambda} = \text{diag}(\lambda_1, \ldots, \lambda_N)$ and $\boldsymbol{\Lambda}_T = \text{diag}(\lambda_{T,1}, \ldots, \lambda_{T,T})$ the corresponding eigenvalues. From (1), the eigendecomposition of the product graph GSO is $\mathbf{S}_{\diamond} = \mathbf{V}_{\diamond}\boldsymbol{\Lambda}_{\diamond}\mathbf{V}_{\diamond}^{\mathsf{H}}$ with eigenvectors $\mathbf{V}_{\diamond} = \mathbf{V}_T \otimes \mathbf{V}$ being the Kronecker product $\otimes$ of the respective GSOs and the eigenvalues $\boldsymbol{\Lambda}_{\diamond} = \boldsymbol{\Lambda}_T \diamond \boldsymbol{\Lambda}$ are defined by the product graph rule. As in graph signal processing [34], it is possible to characterize the joint graph-time Fourier transform of product graph signals. Specifically, the graph-time Fourier transform of signal $\mathbf{x}_{\diamond}$ is defined as $\tilde{\mathbf{x}} = (\mathbf{V}_T \otimes \mathbf{V})^{\mathsf{H}}\mathbf{x}_{\diamond}$ and the eigenvalues in $\boldsymbol{\Lambda}_{\diamond}$ now collect the graph-time frequencies of the product graph [35]. Applying this Fourier transform on the input and output of the GTConv filter in (2), we can write the filter input-output as $\tilde{\mathbf{y}}_{\diamond} = \mathbf{H}(\boldsymbol{\Lambda}_{\diamond})\tilde{\mathbf{y}}$, where $\tilde{\mathbf{y}}_{\diamond}$ is the Fourier transform of the output and $\mathbf{H}(\boldsymbol{\Lambda}_{\diamond})$ is an $NT \times NT$ diagonal matrix containing the filter *frequency response* on the main diagonal. This frequency response is of the form

$$h(\lambda_{\diamond,(n,t)}) = \sum_{k=0}^{K} h_k \lambda_{\diamond,(n,t)}^k \tag{14}$$

where $\lambda_{\diamond,(n,t)} = \lambda_{T,t} \diamond \lambda_n$ indicates the eigenvalue of $\mathbf{S}_{\diamond}$ corresponding to the spatial index $n \in [N]$ and temporal index $t \in [T]$ of the product graph.

The eigenvalues $\lambda_{\diamond,(n,t)}$ can be considered as the frequencies of the product graph and can be ordered in ascending order of magnitude. We can then characterize the variation of the filter frequency response for two different spatial eigenvalues.

**Definition 1.** *A GTConv filter with a frequency response $h(\lambda_{\diamond,(n,t)})$ is graph integral Lipschitz if there exists constant $C > 0$ such that for all frequencies $\lambda_{\diamond,(n,t)}, \lambda_{\diamond,(n',t')} \in \boldsymbol{\Lambda}_{\diamond}$, it holds that*

$$|h(\lambda_{\diamond,(n,t)}) - h(\lambda_{\diamond,(n',t')})| \leq C \frac{|\lambda_n - \lambda_{n'}|}{|\lambda_n + \lambda_{n'}|/2} \quad \text{for all} \quad \{\lambda_n, \lambda_{n'}\} \in \boldsymbol{\Lambda}. \tag{15}$$

Expression (15) states that the frequency response of a GTConv filter should vary sub-linearly while the Lipschitz coefficient $C$ depends on the gap $|\lambda_{\diamond,(n,t)} + \lambda_{\diamond,(n',t')}|/2$. This implies

$$\left| \lambda_n \frac{\partial h(\lambda_{\diamond,(n,t)})}{\partial \lambda_n} \right| \leq C \quad \text{for all} \quad \lambda_n \in \boldsymbol{\Lambda} \quad and \quad \lambda_{\diamond,(n,t)} \in \boldsymbol{\Lambda}_{\diamond} \tag{16}$$

which means the integral Lipschitz filter cannot vary drastically in high graph frequencies. Hence, such a filter can discriminate low frequency content but not high frequency ones.

**Definition 2.** *A GTConv filter has normalized frequency response if $|h(\lambda_{\diamond,(n,t)})| \leq 1$ for all $\lambda_{\diamond,(n,t)}$.*

This definition is a direct consequence of normalizing the filters' frequency response by their maximum value. We shall show next that GTConvAE with filters satisfying Def. 1 and 2 are stable to perturbations in the form (13).

## 3.2 Stability result

The following theorem with proof in Appendix A provides the main result.

**Theorem 1.** *Consider a GTConvAE with an $L_e$-layer encoder and an $L_d$-layer decoder having $F_\ell \leq F_{max}$ and $F_{d,\ell} \leq F_{max}$ features per layer in encoder and decoder, respectively, and a summary function $SUM(\cdot)$ performing pure downsampling with rate $r$. Consider also the filters are integral Lipschitz [cf. Def. 1] with a normalized frequency response [cf. Def. 2] and that the nonlinearities are $1-$Lipschitz (e.g., ReLU, absolute value). Let this GTConvAE be trained over the product graph (1) and deployed over its perturbed version whose spatial GSO is given in (13) with a perturbation of at most $\|\mathbf{E}\| \leq \epsilon$. The distance between the two models is upper bounded by*

$$\|\text{GTConvAE}(\mathbf{x}_\diamond, \mathbf{S}, \mathbf{S}_T) - \text{GTConvAE}(\mathbf{x}_\diamond, \hat{\mathbf{S}}, \mathbf{S}_T)\|_2 \leq (L_d + L_e)r^{-L_e/2}\epsilon\Delta F_{max}^{L_e+L_d-1}\|\mathbf{x}_\diamond\|_2. \tag{17}$$

*where $\Delta = 2C(s_{01} + s_{11}\lambda_{T,max})(1 + \delta\sqrt{NT})$, and $\delta = (\|\mathbf{U} - \mathbf{V}\|^2 + 1)^2 - 1$ with eigenvectors $\mathbf{U}$ from $\mathbf{E} = \mathbf{U}\mathbf{M}\mathbf{U}^\mathsf{H}$ and $\mathbf{V}$ from $\mathbf{S} = \mathbf{V}\mathbf{\Lambda}\mathbf{V}^\mathsf{H}$.*

The result (17) states that GTConvAE is stable against relative perturbations. It also suggests that GTConvAE is less stable for larger product graphs ($\sqrt{NT}$) since more nodes pass information over the perturbed edges. Moreover, making the model more complex by increasing the number of features or layers compromises stability as more graph-time convolutional filters work on a perturbed graph ($F_{max}^{L_e+L_d-1}$). We also see the stability improves with the sampling rate $r > 1$ because fewer nodes operate over the perturbed graph after downsampling. Furthermore, for a deeper encoder we have more downsampling hence the stability improves; yet there is a tradeoff between improving the bound imposed by the terms $r^{-L_e/2}$, $F_{max}^{L_e+L_d-1}$, and $L_e + L_d$. Finally, parameters $s_{01}$ and $s_{11}$ appear in the stability bound because they are the only ones composing the spatial edges; thus, minimizing $\|\mathbf{s}\|_1$ in (12) leads to improved stability.

## 4 Numerical Results

This section compares the GTConvAE with baseline solutions and competitive alternatives for time series denoising as well as anomaly detection with real data from solar irradiance and water networks. In all experiments, the ADAM optimizer with the standard hyperparameters is used and an unweighted directed line graph is considered for the temporal graph in (1).

### 4.1 Denoising of solar irradiance time series

We consider the task of denoising solar irradiance time series over $N = 75$ solar stations around the northern region of the U.S. measured in GHI ($W/m^2$) [4]. Each solar station is a vertex and an undirected edge is set using the physical distances between the stations via Gaussian threshold kernel with $\sigma = 0.25$ and $th = 0.1$ after normalizing maximum weight to 1 [34]. The noise is generated via a zero-mean Gaussian distribution with a covariance matrix corresponding to the pseudo-inverse of the normalized graph Laplacian. More information about the noise model is provided in Appendix B.

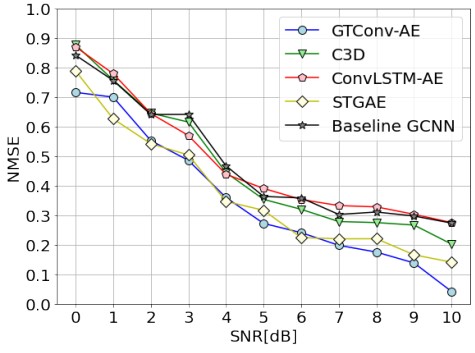

**Figure 2:** Denoising performance of the proposed GTConvAE and alternatives. The standard deviation for all the models is of order $10^{-2}$.

**Experimental setup.** We considered the first 2000 samples for training and validation (2000-2014) and the subsequent 200 (2014-2016) for testing. The input data is a single feature corresponding to the GHI measurement and the product graph has $N = 75$ spatial nodes and $T = 8$ temporal nodes. The GTConvAE has three layers with $\{8, 4, 2\}$ features in the encoder and reversely in the decoder; all filters are 4th-order and normalized Laplacian is used as GSO; a downsampling rate of $r = 2$; a max function in (4); and ReLU activation functions. The regularizer weight in (12) is $\rho = 0.2$ and the learning rate is $25 \times 10^{-4}$. We compared the GTConvAE with the following alternatives:

- *C3D [6]*: non-graph spatiotemporal autoencoder using three-dimensional CNNs.

- *ConvLSTMAE [8]*: A non-graph spatiotemporal autoencoder using two-dimensional CNNs followed by LSTMs.
- *STGAE [1]*: A modular spatiotemporal graph autoencoder that uses an edge varying filter for the graph dimension followed by temporal convolution.
- *Baseline GCNN [46]*: An autoencoder built with a conventional graph convolutional neural network using the time series as features over the nodes. The shift operator is the normalized Laplacian matrix.

The first two methods are considered to show the role of using a distance graph as an inductive bias. The third method is considered to compare the joint GTConvAE over disjoint alternatives, whereas the last model is considered to show the role of the sparse product graphs rather than treating time series as node features. The parameters for all models are chosen via grid search from the ranges reported in Appendix B.

**Results.** Fig. 2 shows the reconstruction normalized mean squared error (NMSE) for different signal-to-noise ratios (SNRs). The proposed GTConvAE compares well with STGAE for low SNRs but better for high SNRs. We attribute this improvement to the ability of the GTConvAE to capture jointly the spatiotemporal patterns in the data while STGAE operates disjointly. We also see that in comparison with the baseline GCNN, the GTConvAE performs consistently better, highlighting the importance of the sparser product graphs and temporal downsampling. Finally, we also observe a superior performance compared with the non-graph alternatives C3D and ConvLSTMAE.

## 4.2 Anomaly detection in water networks

We now consider the task of detecting cyber-physical attacks on a water network. We considered the C-town network from the Battle of ATtack Detection ALgorithms (BATADAL) dataset comprising $N = 388$ nodes (demand junctions, storage tanks, and reservoirs) and 8762 hourly measurements of 43 different node feature signals for a period of 12 months. We used the same setup as in [50] and considered a correlation graph from the data. The dataset provides a normal operating condition comprising recordings for the first 12 months and an anomalous event operating condition comprising 7 attacks over the successive 3 months. Refer to [51, 52] for more detail about the BATADAL dataset.

**Experimental setup.** The normal operating condition data are used to train the model for one-step forecasting to be used for detecting anomalies. The anomalous event operating condition data are used for testing and an anomaly is flagged if the prediction error exceeds a threshold. We set the threshold intuitively to three times the error variance during training. The inputs are the 43 time series over the $N = 388$ nodes and we considered $T = 6$ for the temporal graph dimension. The GTConvAE has two layers with $\{8, 2\}$ features in the encoder and reversely in the decoder; all filters are of order $K = 4$; a downsampling rate $r = 2$; a max function in (4); and ReLU activation functions. The regularizer weight in (12) is $\rho = 0.14$ and learning rate is $5 \times 10^{-4}$. We compared the performance against two graph-based alternatives:

- *STGCAE-LSTM [2]*: A related solution to our method that uses a Cartesian spatiotemporal graph with graph convolutions followed by an LSTM in the latent domain.
- *TGCN [50]*: A modular graph-based autoencoder using cascades of temporal convolutions and message passing.

The parameters for all models are obtained via grid search from the ranges reported in Appendix C. We measure the performance via the $\mathcal{S}$-score present in the BATADAL dataset, which contains $\mathcal{S}_{\text{TTD}}$ for the timing in detecting anomalies and $\mathcal{S}_{\text{CM}}$ for the classification accuracy. The $\mathcal{S}$-score is

$$\mathcal{S} = 0.5(\mathcal{S}_{\text{TTD}} + \mathcal{S}_{\text{CM}}) = 0.5\left( (1 - \frac{1}{N_A} \sum_{i=1}^{N_A} \frac{\text{TTD}_i}{\Delta \text{T}_i}) + \frac{\text{TPR+TNR}}{2} \right), \qquad (18)$$

where $N_A$ is the number of attacks, TTD is the detection time of the attack, $\Delta T_i$ is the duration of the $i-$th attack, TPR is the true positive rate, and TNR is the true negative rate.

**Results:** Table 1 shows that all the models managed to detect all of the attacks, however, the TGCN has a better performance in timing $\mathcal{S}_{\text{TTD}}$. This is due to the calibration of the threshold in their work with a validation dataset while we used a fixed intuitive threshold only based on training. In the

**Table 1:** Comparison of different models in the BATADAL dataset. All metrics are the higher the better.

| Model | $N_A$ | $\mathcal{S}$ | $\mathcal{S}_{\mathrm{TTD}}$ | $\mathcal{S}_{\mathrm{CM}}$ | TPR | TNR |
|---|---|---|---|---|---|---|
| STGCAE-LSTM [2] | 7 | 0.924 | 0.920 | 0.928 | 0.892 | 0.964 |
| TGCN [50] | 7 | 0.931 | 0.934 | 0.928 | 0.885 | 0.971 |
| **GTConvAE** (ours) | 7 | 0.940 | 0.928 | 0.952 | 0.922 | 0.981 |

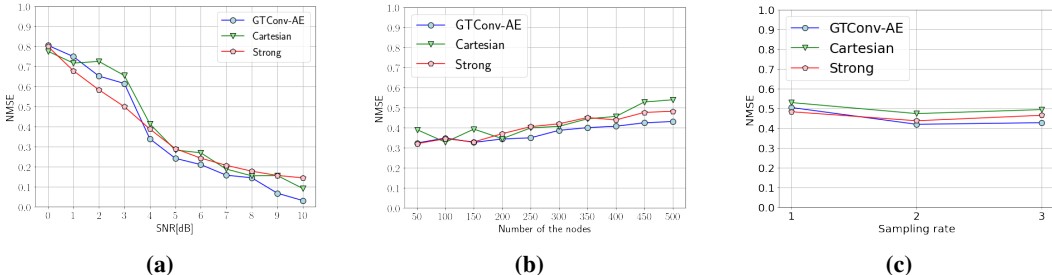

|(a)|(b)|(c)|

**Figure 3:** Stability results for different scenarios of the GTConvAE and fixed product graphs. (a) Different SNRs in the topology. (b) Different graph sizes in 4dB perturbation. (c) Different sampling rates $r$. The standard deviation for all models is of order $\mathcal{O}(10^{-3})$ and it is neglected to avoid overcrowded plots.

accuracy of anomaly detection $\mathcal{S}_{\mathrm{CM}}$, the GTConvAE outperforms the other two models as the product graphs alongside downsampling enable it to learn spatiotemporal patterns in the data effectively. Overall, the GTConvAE performs better than other models by a small margin.

### 4.3 Stability analysis

To investigate the stability of the GTConvAE, we trained the model over a synthesized dataset so we could control all the setting such as the spatial graph size $N$. The graph is an undirected stochastic block model with 5 communities among $N = \{50, 100, \ldots, 500\}$. The edges are drawn independently with probability 0.8 for nodes in the same community and 0.2 otherwise. Each data sample is a diffused signal over the graph $\mathbf{X} = [\mathbf{Sx}, \ldots, \mathbf{S}^T\mathbf{x}]$ with $T = 6$ and $\mathbf{x}$ having a random non-zero entry. The autoencoder is used to reconstruct this data.

**Experimental setup** The model has two layers of encoder and decoder with sampling rate $r = 2$. Each layer of the encoder has $\{8, 4\}$ features and reversely in the decoder. All filters are of order four and the normalized graph Laplacian is used as GSO. The activation functions are ReLU and pure donwsampling is considered. The regularizer weight is $0.25$ and learning rate is $25 \times 10^{-3}$. The model is trained over the graph with different sizes and tested with a perturbed graph following the relative perturbation model in (13) for different SNR scenarios in the topology. We compare the stability of the GTConvAE with learned graphs with the same autoencoder having fixed Cartesian and strong product graphs.

**Results** Fig. 3a indicates that the GTConvAE in different noisy scenarios. GTConvAE is the most stable in medium and high SNRs as it leverages sparsity in the spatiotemporal coupling. However, GTConvAE performance drops more rapidly in low SNR scenarios as its parameters are trained for the data and task. Fig. 3b shows the results for reconstruction error over graphs with different sizes. The GTConvAE is more stable than the other models, even in graphs with the larger sizes for the same reason as before. All models loose performance similarly as the size of the graph grows. This is consistent with the theoretical result in (17).

**Table 2:** Ablation study of GTConvAE over different tasks of denoising, anomaly detection, and data reconstruction.

| model | Denoising (NMSE) | Anomaly detection ($\mathcal{S}$) | Reconstruction (NMSE) | Run time (s/epoch) |
|---|---|---|---|---|
| GTConvAE | 0.28 | 0.940 | 0.24 | 177 |
| Baseline GCNN | 0.36 | 0.889 | 0.35 | 132 |
| Cartesian | 0.31 | 0.913 | 0.29 | 169 |
| Kronecker | 0.32 | 0.908 | 0.28 | 168 |
| Strong | 0.31 | 0.912 | 0.30 | 194 |
| w/o Downsampling | 0.30 | 0.915 | 0.25 | 211 |

## 4.4 Ablation study

To accent the role of each component in GTConvAE, an ablation study has been performed for all previous experiments. The reduced variants of GTConvAE are:

- *Baseline GCNN*: The product graph is removed and time series are represented as distinct features over the nodes of spatial graph.
- *Fixed GTConvAE*: The product graph type is fixed as Cartesian, Kronecker, and Strong product.
- *without Downsampling*: The downsampling module in the encoder and the upsampling module in the decoder are eliminated.

The same experimental setup is considered in each application while the SNR is set to 5 dB in denoising solar irradiance time series. The number of nodes is 250 and sampling rate is 2 for the reconstruction task on synthetic data, and the average run time is provided over 100 different realizations and 100 epochs per experiment. The hardware used to exploit the rune time numbers is a GeForce RTX 3060 GPU, and the code is not optimized for a faster implementation point of view; however, the numbers are used to better comprehend the components' roles.

Table 2 presents the performed ablation study. For all the experiments, GTConvAE outperforms Baseline GCNN due to the product graph and considering time dependencies in the data alongside spatial dependencies. However, the performance improvement comes with increased model complexity as GTConvAE deals with larger graphs while baseline GCNN benefits from a parallel scheme. It is not conclusive which fixed product graph performs better as the optimum product graph may vary depending on the task. Hence, using a parametric product graph in the model can improve the performance, as suggested by these results. Learning through the product graph increases slightly the model complexity compared with the Cartesian and Kronecker models. The downsampling module does not contribute to the performance as long as the data contains smooth time series with short-term patterns; however, it plays a role when it comes to data with longer-term patterns, such as anomaly detection in the water networks. Moreover, it reduces the model complexity significantly and enables GTConvAE to engage on larger datasets.

## 5 Conclusion

We introduced GTConvAE as an unsupervised model for learning representations from multivariate time series over networks. The GTConvAE uses parametric product graphs to aggregate information from a spatiotemporal neighborhood while it still learns spatiotemporal couplings in the product graph. We proposed a spectral analysis for GTConvAE due to its convolutional nature which led to stability analysis. The stability analysis states that GTConvAE is stable against relative perturbations in the spatial graph as long as graph-time filters vary smoothly over high spatiotemporal frequencies. Finally, numerical results showed that the GTConvAE compares well with the state-of-the-art models on benchmark datasets and corroborated the stability results. For further researches, dynamic graphs can be accommodated in the model where the product graph consists of dynamic spatial structures and spatiotemporal edges to tackle more complicated real-world problems.

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

# A  Stability proof

The proof is structured in three components. First we prove the graph-time convolutional filter is stable to perturbations. Then, we prove stability for the encoder and finally for the decoder. Throughout the proof we will use the following lemmas.

**Lemma 1.** *[29] Let* $\mathbf{S} = \mathbf{V}\mathbf{\Lambda}\mathbf{V}^{\mathsf{H}}$ *and* $\mathbf{E} = \mathbf{U}\mathbf{M}\mathbf{U}^{\mathsf{H}}$ *such that* $\|\mathbf{E}\| \leq \epsilon$. *Assume that* $\mathbf{E}_V = \mathbf{V}\mathbf{M}\mathbf{V}^{\mathsf{H}}$ *is the projection of perturbation* $\mathbf{E}$ *over graph eigenspace of* $\mathbf{S}$, *and* $\mathbf{E} = \mathbf{E}_V + \mathbf{E}_U$. *For any eigenvector* $\mathbf{v}_n$ *of* $\mathbf{S}$ *it holds that*

$$\mathbf{E}\mathbf{v}_n = m_n\mathbf{v}_n + \mathbf{E}_U\mathbf{v}_n \tag{19}$$

*with* $\|\mathbf{E}_U\| \leq \epsilon\delta$, *where* $\delta = (\|\mathbf{U} - \mathbf{V}\|^2 + 1)^2 - 1$ *and* $m_n$ *is the* $n$-*th eigenvalue of* $\mathbf{M}$. *Recall that* $\|\cdot\|$ *represents the operator norm of a matrix.*

**Lemma 2.** *Given the frequency response of a graph-time convolutional filter as* $h(\lambda_\diamond) = \sum_{k=1}^{K} h_k\lambda_\diamond^k$, *the partial derivation w.r.t. graph frequency* $\lambda$ *is*

$$\frac{\partial h(\lambda_\diamond)}{\partial \lambda} = (s_{01} + s_{11}\lambda_T)\sum_{k=1}^{K} kh_k\lambda_\diamond^{k-1}. \tag{20}$$

*Proof.* Using the product graph definition (1) we have

$$\frac{\partial \lambda_\diamond}{\partial \lambda} = \frac{\partial(s_{00} + s_{01}\lambda + s_{10}\lambda_T + s_{11}\lambda_T\lambda)}{\partial \lambda} = s_{01} + s_{11}\lambda_T. \tag{21}$$

Then,

$$\frac{\partial h(\lambda_\diamond)}{\partial \lambda} = \frac{\partial h(\lambda_\diamond)}{\partial \lambda_\diamond} \times \frac{\partial \lambda_\diamond}{\partial \lambda} = (\sum_{k=1}^{K} kh_k\lambda_\diamond^{k-1})(s_{01} + s_{11}\lambda_T) \tag{22}$$

completes the proof. $\square$

To ease notation, let us also rearrange the parametric product graph GSO as

$$\mathbf{S}_\diamond = (s_{00}\mathbf{I}_T + s_{10}\mathbf{S}_T) \otimes \mathbf{I}_N + (s_{01}\mathbf{I}_T + s_{11}\mathbf{S}_T) \otimes \mathbf{S} = \mathbf{S}_{T0} \otimes \mathbf{I}_N + \mathbf{S}_{T1} \otimes \mathbf{S} \tag{23}$$

where $\mathbf{S}_{T0} = s_{00}\mathbf{I}_T + s_{10}\mathbf{S}_T$ collects the fully temporal edges and $\mathbf{S}_{T1} = s_{01}\mathbf{I}_T + s_{11}\mathbf{S}_T$ the edges ruled by the spatial graph.

**GTConv filter stability.**

The difference of the filter operating on the perturbed and nominal graph is

$$\mathbf{H}(\mathbf{S}_\diamond) - \mathbf{H}(\hat{\mathbf{S}}_\diamond) = \sum_{k=0}^{K} h_k(\hat{\mathbf{S}}_\diamond^k - \mathbf{S}_\diamond^k) \tag{24}$$

Leveraging the product GSO expansion (23) and the perturbation model $\hat{\mathbf{S}} = \mathbf{S} + (\mathbf{SE} + \mathbf{ES})$ [cf. (13)] we can write the $k$−th power of the perturbed product graph GSO as

$$\begin{aligned}
\hat{\mathbf{S}}_\diamond^k &= (\mathbf{S}_{T0} \otimes \mathbf{I}_N + \mathbf{S}_{T1} \otimes (\mathbf{S} + (\mathbf{SE} + \mathbf{ES})))^k \\
&= (\mathbf{S}_\diamond + (\mathbf{S}_{T1} \otimes (\mathbf{SE} + \mathbf{ES})))^k \\
&= \mathbf{S}_\diamond^k + \sum_{r=0}^{k-1} \mathbf{S}_\diamond^r(\mathbf{S}_{T1} \otimes (\mathbf{SE} + \mathbf{ES}))\mathbf{S}_\diamond^{k-r-1} + \mathbf{D},
\end{aligned} \tag{25}$$

where we applied the first-order Taylor expansion in the third line. Matrix $\mathbf{D}$ contains all terms of order $\mathcal{O}(\epsilon^2)$ and can be ignored.

Substituting then (25) into (24), we get

$$\mathbf{H}(\mathbf{S}_\diamond) - \mathbf{H}(\hat{\mathbf{S}}_\diamond) = \sum_{k=0}^{K} h_k \sum_{r=0}^{k-1} \mathbf{S}_\diamond^r(\mathbf{S}_{T1} \otimes (\mathbf{SE} + \mathbf{ES}))\mathbf{S}_\diamond^{k-r-1}. \tag{26}$$

Upon applying the filters to an input $\mathbf{x}_\diamond$ we get the output difference $\mathbf{y}_\diamond - \hat{\mathbf{y}}_\diamond = (\mathbf{H}(\mathbf{S}_\diamond) - \mathbf{H}(\hat{\mathbf{S}}_\diamond))\mathbf{x}_\diamond$. Substituting into this the graph-time Fourier expansion of the input

$$\mathbf{x}_\diamond = \sum_{t=1}^{T}\sum_{n=1}^{N} \tilde{x}_{(n,t)}(\mathbf{v}_{T,t} \otimes \mathbf{v}_n) \tag{27}$$

with $\tilde{x}_{(n,t)}$ the $(n,t)-$th Fourier coefficients and $(\mathbf{v}_{T,t}, \mathbf{v}_n)$ the eigenvector pair for the temporal and spatial GSOs [cf. Sec. 3.1], we can write the output difference as

$$\mathbf{y}_\diamond - \hat{\mathbf{y}}_\diamond = \sum_{t=1}^{T}\sum_{n=1}^{N} \tilde{x}_{(n,t)} \sum_{k=0}^{K} h_k \sum_{r=0}^{k-1} \mathbf{S}_\diamond^r (\mathbf{S}_{T1} \otimes (\mathbf{SE}+\mathbf{ES}))\mathbf{S}_\diamond^{k-r-1}(\mathbf{v}_{T,t} \otimes \mathbf{v}_n). \tag{28}$$

Since $(\mathbf{v}_{T,t} \otimes \mathbf{v}_n)$ is an eigenvector of $\mathbf{S}_\diamond$, we have

$$\mathbf{S}_\diamond^{k-r-1}(\mathbf{v}_{T,t} \otimes \mathbf{v}_n) = \lambda_{\diamond,(n,t)}^{k-r-1}(\mathbf{v}_{T,t} \otimes \mathbf{v}_n) \tag{29}$$

which by substituting to (28) yields

$$\mathbf{y}_\diamond - \hat{\mathbf{y}}_\diamond = \sum_{t=1}^{T}\sum_{n=1}^{N} \tilde{x}_{(n,t)} \sum_{k=0}^{K} h_k \sum_{r=0}^{k-1} \lambda_{\diamond,(n,t)}^{k-r-1}\mathbf{S}_\diamond^r(\mathbf{S}_{T1} \otimes (\mathbf{SE}+\mathbf{ES}))(\mathbf{v}_{T,t} \otimes \mathbf{v}_n) \tag{30}$$

where $\lambda_{\diamond,(n,t)}$ is the eigenvalue of the product graph GSO $\mathbf{S}_\diamond$ for indices $(n,t)$. Leveraging mixed product property of Kronecker product[2] allows us to rewrite (30) as

$$\mathbf{y}_\diamond - \hat{\mathbf{y}}_\diamond = \sum_{t=1}^{T}\sum_{n=1}^{N} \tilde{x}_{(n,t)} \sum_{k=0}^{K} h_k \sum_{r=0}^{k-1} \lambda_{\diamond,(n,t)}^{k-r-1}\mathbf{S}_\diamond^r(\mathbf{S}_{T1}\mathbf{v}_{T,t} \otimes (\mathbf{SE}+\mathbf{ES})\mathbf{v}_n). \tag{31}$$

Replacing $\mathbf{S}_{T1} = s_{01}\mathbf{I}_T + s_{11}\mathbf{S}_T$ leads to

$$\hat{\mathbf{y}}_\diamond - \mathbf{y}_\diamond = \sum_{t=1}^{T}\sum_{n=1}^{N} (s_{01}+s_{11}\lambda_{T,t})\tilde{x}_{(n,t)} \sum_{k=0}^{K} h_k \sum_{r=0}^{k-1} \lambda_{\diamond,(n,t)}^{k-r-1}\mathbf{S}_\diamond^r(\mathbf{v}_{T,t} \otimes (\mathbf{SE}+\mathbf{ES})\mathbf{v}_n). \tag{32}$$

Applying Lemma 1 results in

$$\hat{\mathbf{y}}_\diamond - \mathbf{y}_\diamond = \sum_{t=1}^{T}\sum_{n=1}^{N} (s_{01}+s_{11}\lambda_{T,t})\tilde{x}_{(n,t)} \sum_{k=0}^{K} h_k \sum_{r=0}^{k-1} \lambda_{\diamond,(n,t)}^{k-r-1}\mathbf{S}_\diamond^r(\mathbf{v}_{T,t} \otimes (\mathbf{S}+\lambda_n\mathbf{I}_N)(\underbrace{m_n\mathbf{v}_n}_{\text{term 1}}+\underbrace{\mathbf{E}_U\mathbf{v}_n}_{\text{term 2}})), \tag{33}$$

which leaves us with two terms that shall be discussed separately.

For the first term, we have

$$\mathbf{t}_1 = \sum_{t=1}^{T}\sum_{n=1}^{N} 2\lambda_n m_n(s_{01}+s_{11}\lambda_{T,t})\tilde{x}_{(n,t)} \sum_{k=0}^{K} h_k \sum_{r=0}^{k-1} \lambda_{\diamond,(n,t)}^{k-r-1}\mathbf{S}_\diamond^r(\mathbf{v}_{T,t} \otimes \mathbf{v}_n). \tag{34}$$

By exploiting eigenvector property $\mathbf{S}_\diamond^r(\mathbf{v}_{T,t} \otimes \mathbf{v}_n) = \lambda_{\diamond,(n,t)}^r(\mathbf{v}_{T,t} \otimes \mathbf{v}_n)$ we can rewrite (34) into

$$\mathbf{t}_1 = \sum_{t=1}^{T}\sum_{n=1}^{N} 2\lambda_n m_n(s_{01}+s_{11}\lambda_{T,t})\tilde{x}_{(n,t)} \sum_{k=0}^{K} k h_k \lambda_{\diamond,(n,t)}^{k-1}(\mathbf{v}_{T,t} \otimes \mathbf{v}_n). \tag{35}$$

Applying Lemma 2 leads to

$$\mathbf{t}_1 = \sum_{t=1}^{T}\sum_{n=1}^{N} 2m_n\tilde{x}_{(n,t)}\lambda_n \frac{\partial h(\lambda_{\diamond,(n,t)})}{\partial\lambda_n}(\mathbf{v}_{T,t} \otimes \mathbf{v}_n). \tag{36}$$

For the second term, we have

$$\mathbf{t}_2 = \sum_{t=1}^{T}\sum_{n=1}^{N} (s_{01}+s_{11}\lambda_{T,t})\tilde{x}_{(n,t)} \sum_{k=0}^{K} h_k \sum_{r=0}^{k-1} \lambda_{\diamond,(n,t)}^{k-r-1}\mathbf{S}_\diamond^r(\mathbf{v}_{T,t} \otimes (\mathbf{S}+\lambda_n\mathbf{I}_N)\mathbf{E}_U\mathbf{v}_n). \tag{37}$$

---
[2] $(A \otimes B)(C \otimes D) = AC \otimes BD$

By substituting the eigendecomposition $\mathbf{S}_\diamond^r = (\mathbf{V}_T \otimes \mathbf{V}) \mathbf{\Lambda}_\diamond^r (\mathbf{V}_T \otimes \mathbf{V})^{\mathsf{H}}$ we get

$$\mathbf{t}_2 = \sum_{t=1}^{T} \sum_{n=1}^{N} \tilde{x}_{(n,t)} (\mathbf{V}_T \otimes \mathbf{V}) \mathrm{diag}(\mathbf{g}_{(n,t)}) (\mathbf{V}_T \otimes \mathbf{V})^{\mathsf{H}} (\mathbf{v}_{T,t} \otimes \mathbf{E}_U \mathbf{v}_n). \tag{38}$$

where the entries of vectors $\mathbf{g}_{(n,t)} \in \mathbb{R}^{NT}$ for $n \in [N]$ and $t \in [T]$ are defined as

$$
\begin{aligned}
g_{(n,t)}(n', t') &= (s_{01} + s_{11}\lambda_{T,t})(\lambda_n + \lambda_{n'}) \sum_{k=0}^{k} h_k \sum_{r=0}^{k-1} \lambda_{\diamond,(n,t)}^{k-r-1} \lambda_{\diamond,(n',t')}^{r} \\
&= \begin{cases} 2\lambda_n \dfrac{\partial h(\lambda_{\diamond,(n,t)})}{\partial \lambda_n}; & (n,t) = (n',t') \\[2mm] (s_{01} + s_{11}\lambda_{T,t})(h(\lambda_{\diamond,(n,t)}) - h(\lambda_{\diamond,(n',t')})) \dfrac{\lambda_n + \lambda_{n'}}{\lambda_n - \lambda_{n'}}; & (n,t) \neq (n',t') \end{cases}
\end{aligned} \tag{39}
$$

With this in place, we now upper bound the two-norm of the difference $\mathbf{y}_\diamond - \hat{\mathbf{y}}_\diamond = \mathbf{t}_1 + \mathbf{t}_2$ by bounding each of the terms $\mathbf{t}_1$ and $\mathbf{t}_2$ separately. From $\|\mathbf{E}\| \leq \epsilon$, we have that $|m_n| \leq \epsilon$. Also from the integral Lipschitz property of the filter [cf. Def. 1]. Using these two into (36), we can upper bound the norm of term $\mathbf{t}_1$ as

$$\|\mathbf{t}_1\|_2 \leq 2\epsilon C \sum_{t=1}^{T} \sum_{n=1}^{N} \tilde{x}_{(n,t)} (\mathbf{v}_{T,t} \otimes \mathbf{v}_n) \leq 2\epsilon C \|\mathbf{x}_\diamond\|_2, \tag{40}$$

where the second inequality holds due to Fourier transform definition (27).

Moving on to $\mathbf{t}_2$, we use mixed product property as $\mathbf{v}_{T,t} \otimes \mathbf{E}_U \mathbf{v}_n = (\mathbf{I}_T \otimes \mathbf{E}_U)(\mathbf{v}_{T,t} \otimes \mathbf{v}_n)$ and operator norms in (38) to obtain an upper bound as

$$\|\mathbf{t}_2\|_2 \leq \sum_{t=1}^{T} \sum_{n=1}^{N} |\tilde{x}_{(n,t)}| \|(\mathbf{V}_T \otimes \mathbf{V})\| \|\mathrm{diag}(\mathbf{g}_{(n,t)})\| \|(\mathbf{V}_T \otimes \mathbf{V})^{\mathsf{H}}\| \|\mathbf{I}_T \otimes \mathbf{E}_U\| \|\mathbf{v}_{T,t} \otimes \mathbf{v}_n\|_2. \tag{41}$$

From the integral Lipschitz property we can bound $\|\mathrm{diag}(\mathbf{g}_{(n,t)})\| \leq 2C(s_{01} + s_{11}\lambda_{T,max})$ in (39) where $\lambda_{T,max}$ is a temporal eigenvalue with the largest absolute value. As $\mathbf{V}_T \otimes \mathbf{V}$ is an orthonormal bases, its operator norm is $\|\mathbf{V}_T \otimes \mathbf{V}\| = 1$, and $l_2$-norm of the eigenvectors is $\|\mathbf{v}_{T,t} \otimes \mathbf{v}_n\|_2 = 1$. Lemma 1 states that $\|\mathbf{E}\| \leq \epsilon\delta$ which leads to $\|\mathbf{I}_T \otimes \mathbf{E}_U\| \leq \epsilon\delta$. Finally, $l_1$-norm can be bounded by $\sum_{t=1}^{T} \sum_{n=1}^{N} |\tilde{x}_{(n,t)}| = \|\tilde{\mathbf{x}}\|_1 \leq \sqrt{NT}\|\tilde{\mathbf{x}}\|_2 = \sqrt{NT}\|\mathbf{x}_\diamond\|_2$. Considering all the abovementioned bounds and replacing them in (41) yields

$$\|\mathbf{t}_2\|_2 \leq 2(s_{01} + s_{11}\lambda_{T,max})\epsilon C \delta \sqrt{NT} \|\mathbf{x}_\diamond\|_2. \tag{42}$$

Finally, based on the triangle inequality the GTConv filter difference is

$$\|\mathbf{H}(\mathbf{S}_\diamond) - \mathbf{H}(\hat{\mathbf{S}}_\diamond)\| \leq 2(s_{01} + s_{11}\lambda_{T,max})\epsilon C(1 + \delta\sqrt{NT}) = \epsilon\Delta. \tag{43}$$

**Encoder stability.**

Consider the encoder contains $L_e$ layer each having $F_\ell$ features and $r$ sampling rate. We are interested in the output difference of the encoder

$$\|\mathrm{ENC}(\mathbf{x}_\diamond, \mathbf{S}, \mathbf{S}_T) - \mathrm{ENC}(\mathbf{x}_\diamond, \hat{\mathbf{S}}, \mathbf{S}_T)\|_2^2 = \sum_{f=1}^{F_{L_e}} \|\mathbf{x}_{\diamond,L_e}^f - \hat{\mathbf{x}}_{\diamond,L_e}^f\|_2^2. \tag{44}$$

To ease exposition, we denote $\mathbf{H} := \mathbf{H}(\mathbf{S})$ and $\hat{\mathbf{H}} := \mathbf{H}(\hat{\mathbf{S}})$. For the $f-$th output encoder feature we have

$$\|\mathbf{x}_{\diamond,L_e}^f - \hat{\mathbf{x}}_{\diamond,L_e}^f\|_2 = \left\| \sigma\left( \sum_{g=1}^{F_{L_e-1}} S_r(\mathbf{H}_{L_e}^{fg} \mathbf{x}_{\diamond,L_e-1}^g) \right) - \sigma\left( \sum_{g=1}^{F_{L_e-1}} S_r(\hat{\mathbf{H}}_{L_e}^{fg} \hat{\mathbf{x}}_{\diamond,L_e-1}^g) \right) \right\|_2 \tag{45}$$

where $S_r(\cdot)$ is the sampling operator with rate $r$, i.e., simple $SUM(\cdot)$ function without any aggregation. The downsampling reduces the norm of each time series by a factor $1/\sqrt{r}$, so $\|\mathbf{y}_{\diamond,L_e}\|_2$ will be

reduced by $1/\sqrt{r}$. As non-linearity is 1-Lipschitz, i.e., $|\sigma(a) - \sigma(b)| \leq |a - b|$, we can conclude the following inequality from (45) by use of triangular inequality

$$\|\mathbf{x}_{\diamond,L_e}^f - \hat{\mathbf{x}}_{\diamond,L_e}^f\|_2 \leq \frac{1}{\sqrt{r}} \sum_{g=1}^{F_{L_e-1}} \left\| \mathbf{H}_{L_e}^{fg} \mathbf{x}_{\diamond,L_e-1}^g - \hat{\mathbf{H}}_{L_e}^{fg} \hat{\mathbf{x}}_{\diamond,L_e-1}^g \right\|_2. \tag{46}$$

We add and subtract $\hat{\mathbf{H}}_L^{fg} \mathbf{x}_{\diamond,L-1}^g$ inside the $l_2$-norm and use the triangular inequality once again for each of the input features $g$ to get

$$\left\| \mathbf{H}_{L_e}^{fg} \mathbf{x}_{\diamond,L_e-1}^g - \hat{\mathbf{H}}_{L_e}^{fg} \hat{\mathbf{x}}_{\diamond,L_e-1}^g \right\|_2 \leq \|(\mathbf{H}_{L_e}^{fg} - \hat{\mathbf{H}}_{L_e}^{fg}) \mathbf{x}_{\diamond,L_e-1}^g\|_2 + \|\hat{\mathbf{H}}_{L_e}^{fg}(\mathbf{x}_{\diamond,L_e-1}^g - \hat{\mathbf{x}}_{\diamond,L_e-1}^g)\|_2$$

$$\leq \|\mathbf{H}_{L_e}^{fg} - \hat{\mathbf{H}}_{L_e}^{fg}\|\|\mathbf{x}_{\diamond,L_e-1}^g\|_2 + \|\hat{\mathbf{H}}_{L_e}^{fg}\|\|\mathbf{x}_{\diamond,L_e-1}^g - \hat{\mathbf{x}}_{\diamond,L_e-1}^g\|_2 \tag{47}$$

The stability of GTConv filter in (43) provides an upper bound for the first term as $\|\mathbf{H}_{L_e}^{fg} - \hat{\mathbf{H}}_{L_e}^{fg}\| \leq \epsilon \Delta$ which is applicable for all the layers. Note that $\Delta$ depends on temporal graph size, so it is different in each layer due to the downsampling. However, we assume the largest temporal size $T$ so the inequality holds for all the layers [3]. The second term is bounded by spectral normalization assumption $\|\mathbf{H}_{L_e}^{fg}\| \leq 1$ [cf. Def. 2]. Leveraging these bounds and replacing in (46) we get

$$\|\mathbf{x}_{\diamond,L_e}^f - \hat{\mathbf{x}}_{\diamond,L_e}^f\|_2 \leq \frac{1}{\sqrt{r}} \sum_{g=1}^{F_{L_e-1}} \epsilon \Delta \|\mathbf{x}_{\diamond,L_e-1}^g\|_2 + \|\mathbf{x}_{\diamond,L_e-1}^g - \hat{\mathbf{x}}_{\diamond,L_e-1}^g\|_2. \tag{48}$$

This equation defines a recursion among the encoder layers with initial condition $\mathbf{x}_{\diamond,0}^g = \hat{\mathbf{x}}_{\diamond,0}^g := \mathbf{x}_{\diamond}^g$ for all the input features. So for the $\ell-$th layer, we can write

$$\|\mathbf{x}_{\diamond,\ell}^f - \hat{\mathbf{x}}_{\diamond,\ell}^f\|_2 \leq \frac{1}{\sqrt{r}} \sum_{g=1}^{F_{\ell-1}} \epsilon \Delta \|\mathbf{x}_{\diamond,\ell-1}^g\|_2 + \|\mathbf{x}_{\diamond,\ell-1}^g - \hat{\mathbf{x}}_{\diamond,\ell-1}^g\|_2. \tag{49}$$

To solve this recursive inequality, we first upper bound $\|\mathbf{x}_{\diamond,\ell}^f\|_2$ as

$$\|\mathbf{x}_{\diamond,\ell}^f\|_2 \leq \frac{1}{\sqrt{r}} \sum_{g=1}^{F_{\ell-1}} \|\mathbf{H}_{\ell}^{fg} \mathbf{x}_{\diamond,\ell-1}^g\|_2 \leq \frac{1}{\sqrt{r}} \sum_{g=1}^{F_{\ell-1}} \|\mathbf{x}_{\diamond,\ell-1}^g\|_2, \tag{50}$$

where the last inequality is due to the assumption $\|\mathbf{H}_{\ell}^{fg}\| \leq 1$ [Def. 2]. Solving this recursion leads to

$$\|\mathbf{x}_{\diamond,\ell}^f\|_2 \leq \frac{1}{r^{l/2}} \prod_{i=1}^{\ell-1} F_i \sum_{g=1}^{F_0} \|\mathbf{x}_{\diamond}^g\|_2 = r^{-\ell/2} \prod_{i=1}^{\ell-1} F_i \sum_{g=1}^{F_0} \|\mathbf{x}_{\diamond}^g\|_2. \tag{51}$$

Replacing (51) in (49) and solving the recursion considering the initial conditions we get

$$\|\mathbf{x}_{\diamond,\ell}^f - \hat{\mathbf{x}}_{\diamond,\ell}^f\|_2 \leq r^{-\ell/2} \epsilon \Delta \ell \prod_{i=1}^{\ell-1} F_i \sum_{g=1}^{F_0} \|\mathbf{x}_{\diamond}^g\|_2. \tag{52}$$

Setting $\ell = L_e$ in (52) and replacing it in (44) yields to

$$\|\text{ENC}(\mathbf{x}_{\diamond}, \mathbf{S}, \mathbf{S}_T) - \text{ENC}(\mathbf{x}_{\diamond}, \hat{\mathbf{S}}, \mathbf{S}_T)\|_F \leq L_e r^{-L_e/2} \epsilon \Delta \sqrt{F_{L_e}} \prod_{n=1}^{L_e-1} F_n \sum_{g=1}^{F_0} \|\mathbf{x}_{\diamond}^g\|_2. \tag{53}$$

**GTConv-AE stability.**

Let $\mathbf{Z}_{\diamond} = \text{ENC}(\mathbf{x}_{\diamond}, \mathbf{S}, \mathbf{S}_T)$ be the input of the decoder and $\mathbf{z}_{\diamond,L_d} = \text{DEC}(\mathbf{Z}_{\diamond}, \mathbf{S}, \mathbf{S}_T)$ its output. To prove GTConvAE stability, we need to bound

$$\|\text{DEC}(\mathbf{Z}_{\diamond}, \mathbf{S}, \mathbf{S}_T) - \text{DEC}(\mathbf{Z}_{\diamond}, \hat{\mathbf{S}}, \mathbf{S}_T)\|_2^2 = \sum_{f=1}^{F_{d,L_d}} \|\mathbf{z}_{\diamond,L_d}^f - \hat{\mathbf{z}}_{\diamond,L_d}^f\|_2^2. \tag{54}$$

---

[3] It is possible to solve the recursive equation with $\Delta_T$ as a variable, but it leads to overcrowded multipliers in inequalities without carrying important information on the bound.

For each feature in the output we have

$$\|\mathbf{z}_{\diamond,L_d}^f - \hat{\mathbf{z}}_{\diamond,L_d}^f\|_2 = \left\| \sigma \left( \sum_{g=1}^{F_{d,L_d-1}} U_r(\mathbf{H}_{L_d}^{fg} \mathbf{z}_{\diamond,L_d-1}^g) \right) - \sigma \left( \sum_{g=1}^{F_{d,L_d-1}} U_r(\hat{\mathbf{H}}_{L_d}^{fg} \hat{\mathbf{z}}_{\diamond,L_d-1}^g) \right) \right\|_2 \quad (55)$$

where $U_r(\cdot)$ is an upsampling operator with rate $r$ which insert zeros among the samples. The upsampling module leaves the $l_2$-norm per time series unaffected and can be ignored. Given 1-Lipschitz continuity of activation function $\sigma(\cdot)$, the following inequality can be concluded from (55) using the triangular inequality

$$\|\mathbf{z}_{\diamond,L_d}^f - \hat{\mathbf{z}}_{\diamond,L_d}^f\|_2 \leq \sum_{g=1}^{F_{d,L_d-1}} \left\| \mathbf{H}_{L_d}^{fg} \mathbf{z}_{\diamond,L_d-1}^g - \hat{\mathbf{H}}_{L_d}^{fg} \hat{\mathbf{z}}_{\diamond,L_d-1}^g \right\|_2. \quad (56)$$

Adding and subtracting $\hat{\mathbf{H}}_{L_d}^{fg} \mathbf{z}_{\diamond,L_d-1}^g$ in the norm and leveraging again the triangular inequality yields

$$\left\| \mathbf{H}_{L_d}^{fg} \mathbf{z}_{\diamond,L_d-1}^g - \hat{\mathbf{H}}_{L_d}^{fg} \hat{\mathbf{z}}_{\diamond,L_d-1}^g \right\|_2 \leq \|(\mathbf{H}_{L_d}^{fg} - \hat{\mathbf{H}}_{L_d}^{fg})\mathbf{z}_{\diamond,L_d-1}^g\|_2 + \|\hat{\mathbf{H}}_{L_d}^{fg}(\mathbf{z}_{\diamond,L_d-1}^g - \hat{\mathbf{z}}_{\diamond,L_d-1}^g)\|_2$$
$$\leq \|\mathbf{H}_{L_d}^{fg} - \hat{\mathbf{H}}_{L_d}^{fg}\|\|\mathbf{z}_{\diamond,L_d-1}^g\|_2 + \|\hat{\mathbf{H}}_{L_d}^{fg}\|\|\mathbf{x}_{\diamond,L_d-1}^g - \hat{\mathbf{z}}_{\diamond,L_d-1}^g\|_2, \quad (57)$$

for $g = 1, \ldots, F_{d,L_d-1}$. The first term is bounded by GTConv filters stability in (43) and the second term is upper-bounded because filters are normalized $\|\mathbf{H}_\ell^{fg}\| \leq 1$ [cf. Def. 2]. Given these two bounds, (57) can be upper-bounded as

$$\|\mathbf{z}_{\diamond,L_d}^f - \hat{\mathbf{z}}_{\diamond,L_d}^f\|_2 \leq \sum_{g=1}^{F_{d,L_d-1}} \epsilon \Delta \|\mathbf{z}_{\diamond,L_d-1}^g\|_2 + \|\mathbf{z}_{\diamond,L_d-1}^g - \hat{\mathbf{z}}_{\diamond,L_d-1}^g\|_2. \quad (58)$$

This allows defining a recursion for the generic layer $\ell$ as

$$\|\mathbf{z}_{\diamond,\ell}^f - \hat{\mathbf{z}}_{\diamond,\ell}^f\|_2 \leq \sum_{g=1}^{F_{d,\ell-1}} \epsilon \Delta \|\mathbf{z}_{\diamond,\ell-1}^g\|_2 + \|\mathbf{z}_{\diamond,\ell-1}^g - \hat{\mathbf{z}}_{\diamond,\ell-1}^g\|_2. \quad (59)$$

For the first term on the right hand-side of (59), we have

$$\|\mathbf{z}_{\diamond,\ell}^f\|_2 \leq \sum_{g=1}^{F_{d,\ell-1}} \|\mathbf{H}_\ell^{fg} \mathbf{z}_{\diamond,\ell-1}^g\|_2 \leq \sum_{g=1}^{F_{d,\ell-1}} \|\mathbf{z}_{\diamond,\ell-1}^g\|_2 = \prod_{j=1}^{\ell-1} F_{d,j} \sum_{g=1}^{F_{d,0}} \|\mathbf{z}_{\diamond,0}^g\|_2 \quad (60)$$

because $\|\mathbf{H}_\ell^{fg}\| \leq 1$ [cf. Def. 2]. Replacing (60) into (59) and evaluating it at $\ell = L_d$ brings the recursion to its initial conditions

$$\|\mathbf{z}_{\diamond,L_d}^f - \hat{\mathbf{z}}_{\diamond,L_d}^f\|_2 \leq \epsilon \Delta L_d \prod_{j=1}^{L_d-1} F_{d,j} \sum_{g=1}^{F_{d,0}} \|\mathbf{z}_{\diamond,0}^g\|_2 + \prod_{j=1}^{L_d-1} F_{d,j} \sum_{g=1}^{F_{d,0}} \|\mathbf{z}_{\diamond,0}^g - \hat{\mathbf{z}}_{\diamond,0}^g\|_2. \quad (61)$$

For initial conditions we have $\mathbf{Z}_{\diamond,0} = \mathbf{Z}_\diamond$, however, the error caused by spatial graph perturbation in the encoder appears here as an initial condition where $\|\mathbf{z}_{\diamond,0}^f - \hat{\mathbf{z}}_{\diamond,0}^f\|_2$ is bounded by the result in (53) for $f \in [F_{d,0}]$.

As the initial condition of the decoder states $\mathbf{Z}_{\diamond,0} = \mathbf{Z}_\diamond = \mathbf{X}_{\diamond,L}$, we can set $\ell = L$ in (51) to obtain

$$\|\mathbf{z}_\diamond^f\|_2 \leq r^{-L_e/2} \prod_{i=1}^{L_e-1} F_i \sum_{g=1}^{F_0} \|\mathbf{x}_\diamond^g\|_2. \quad (62)$$

Substituting encoder stability bound (53), to enforce the initial condition for $\sum_{g=1}^{F_{d,0}} \|\mathbf{z}_{\diamond,0}^g - \hat{\mathbf{z}}_{\diamond,0}^g\|_2$, and (62) into (61) results in

$$\|\mathbf{z}_{\diamond,L_d}^f - \hat{\mathbf{z}}_{\diamond,L_d}^f\|_2 \leq L_d r^{-L_e/2} \epsilon \Delta F_{d,0} \prod_{i=1}^{L_e-1} F_i \prod_{j=1}^{L_d-1} F_{d,j} \sum_{g=1}^{F_0} \|\mathbf{x}_\diamond^g\|_2$$
$$+ L_e r^{-L_e/2} \epsilon \Delta F_{d,0} \prod_{i=1}^{L_e-1} F_i \prod_{j=1}^{L_d-1} F_{d,j} \sum_{g=1}^{F_0} \|\mathbf{x}_\diamond^g\|_2. \quad (63)$$

Calculating over all the output features completes the upper-bound as

$$\|\text{GTConvAE}(\mathbf{x}_\diamond, \mathbf{S}, \mathbf{S}_T) - \text{GTConvAE}(\mathbf{x}_\diamond, \hat{\mathbf{S}}, \mathbf{S}_T)\|_2 \leq$$

$$(L_d + L_e) r^{-L_e/2} \epsilon \Delta \sqrt{F_{d,L_d}} \prod_{i=1}^{L_e-1} F_i \prod_{j=0}^{L_d-1} F_{d,j} \sum_{g=1}^{F_0} \|\mathbf{x}_\diamond^g\|_2. \tag{64}$$

Assuming $F_0 = F_{d,L_d} = 1$ and $\{F_d, F\} \leq F_{max}$ completes the proof. $\qquad\square$

## B Denoising solar irradiance time series

In this appendix we provide extra information on numerical experiment for denoising solar irradiance time series.

**SNR:** An error vector $\mathbf{e}_t \sim \mathcal{N}(0, \mathbf{L}^\dagger)$ is generated independently for each timestamp $t \in [T]$. Matrix $\mathbf{L}$ represents normalized Laplacian and $\dagger$ stands for pseudo-inverse operation. This noise varies smoothly over spatial graph which makes it more difficult to detect. Assume noise matrix $\sigma\mathbf{E} = \sigma[\mathbf{e}_1, \ldots, \mathbf{e}_T] \in \mathbb{R}^{N \times T}$, we define SNR as follow:

$$SNR = 20 \log \frac{\|\mathbf{X}\|_F}{\sigma\|\mathbf{E}\|_F}, \tag{65}$$

where $\sigma$ is used to control SNR for the experiments.

**GTConvAE parameters:** The time window is searched over $T \in \{2, \ldots, 8\}$. The number of layers for both encoder and decoder are selected from $L_e = L_d \in \{2, 3\}$. The number of features for every layer are chosen from $F \in \{32, 16, 8, 4, 2\}$. The filter order is evaluated on $K \in \{2, 3, 4, 5\}$. The sampling is searched over $r \in \{1, 2, 3, 4\}$. All the aggregation function have been tested. Finally, the regularizer weight initially selected from logarithmic interval $\rho \in \{10^-2, \ldots, 10^2\}$ and fine-tuned around optimum value.

**C3D parameters:** As proposed in [6], 4 convolutional layer, 2 subsampling layer, and 2 fully connected layer are considered for both encoder and decoder. The time window is searched over $T \in \{4, \ldots, 6\}$. The number of features for every layer are chosen from $F \in \{32, 16, 8, 4, 2\}$. So, the size of 3D convolution in each layer is $75 \times F \times T$. The number of channels are selected from $C = \{12, 24, 48\}$ in each layer. Fully connected layers have 128 and 64 features, respectively. The learning rate is set to $0.01$.

**ConvLSTM-AE parameters:** The time window $T = 10$ as it advised in [8]. Both encoder and decoder have 3 convolutional layer connected in series with ConvLSTM modules. The number of features for every layer are chosen from $F \in \{32, 16, 8, 4, 2\}$. The design is symmetrically identical in the decoder. The learning rate is set to $0.01$.

**STGAE:** The learning rate is reducing from $0.01$ as the loss function declines. An early stopping mechanism is used to avoid overfitting. Temporal convolution size is selected from $T = \{4, 5, 6\}$. Each graph-based attention convolution holds $F \in \{32, 16, 8, 4, 2\}$ chosen via grid search. Minor parameters are the same as [1].

## C Anomaly detection in water networks

In this appendix we provide extra information on numerical experiments for anomaly detection in water networks.

**GTConvAE parameters:** The model parameters are evaluated and fine-tuned by sliding window backtesting. The time window is searched over $T \in \{2, \ldots, 8\}$. The number of layers for both encoder and decoder are selected from $L_e = L_d \in \{2, 3\}$. The number of features for every layer are chosen from $F \in 128, 64, 32, 16, 8, 4, 2$. The filter order is evaluated on $K \in \{2, 3, 4, 5\}$. The sampling is searched over $r \in \{1, 2, 3\}$. All the aggregation functions have been tested. Finally, the regularizer weight initially selected from logarithmic interval $\rho \in \{10^-2, \ldots, 10^2\}$ and fine-tuned around optimum value.

**STGCAE-LSTM:** The time window is set to 12 with stride 1 in temporal convolution units similar to [2]. The inner LSTM network has 12 layers with features selected from $F \in 128, 64, 32, 16, 8, 4, 2$. The regularization parameter is 0.05. The learning rate is $10^-3$.

**TGCN:** For the complete experimental setup refer to [50].

