# OpenReview forum: "Graph-Time Convolutional Autoencoders"
_logconference.io/LOG/2022/Conference — LoG 2022 Poster_

### Official Review · Reviewer_FWAV · 2022-09-27

**Overall Score:** 3
**Confidence:** 3

**Review:**

This paper proposes graph-time convolutional autoencoders (GTConvAE) for unsupervised learning on the temporal graph domain. The paper also provides some stability analysis for the proposed model. Experiments on solar and water networks show the empirical performance of the proposed model compared to three baselines.

[+] The targeting problem of spatio-temporal graph unsupervised learning seems new and interesting, especially in nature science domains, such as solar and water systems.

[+] The reviewer did not check the equation formulation thoroughly; however, it appears sound.

[-] The major concern of the proposed method is the space and time complexity. Although the paper has analyzed the complexity in Section 2, reporting the space and time comparison against baseline methods is strongly recommended.

[-] The model development needs more motivation. The novelty of the proposed model should be highlighted.

[-] The experiments could be stronger with ablation studies for providing more insights, such as how the temporal information helps, how the graph structure information helps, and will some small perturbation on graph structure affects the model? To what extent?

Minor:
- Related literature could be better organized
- The proposed method might be compared to basic ML methods, such as basic autoencoder and logistic regression.

---

### Official Review · Reviewer_bYcN · 2022-10-18

**Overall Score:** 6
**Confidence:** 3

**Review:**

This paper addresses the problem of defining architectures for graph-time convolutional autoencoders.
The key point of the article is the definition of (1), where a GSO is represented by means of product graphs, yielding a representation of the dynamic data on the graph as a static signal of the product graph. The other options chosen by the authors for temporal downsampling and upsampling, activation functions, and losses seem pretty natural to me.
The proposition in (1) is supported by the fact that it is stable in the sense of [27]. A formal mathematical proof is included in the annex.
Experimental section shows the goodness of the author’s proposition for denoising solar irradiance time series (using the decoder output as denoiser) and anomaly detection in water networks.

This reviewer thinks that the weakest part of the article is the lack of balance between the theoretical results (which this reviewer considers very good) and the interpretation of the results (which seems quite limited to me).

There are some aspects that I find interesting to improve the document:

Could you please explain the numerically complexity of (1).

Are the values of {sij} interpretable?

Finally, in many problems it is important to use augmentation methods to avoid overlearning. Is stability a property that allows knowing what type of augmentation should not be used?

Accordingly,
This reviewer thinks the article has the merit to be considered for the conference. My score is "weak accepted"

Style recommendations:

Line 28:  LTSM networks -> LSTM networks  (please define acronym)

Line113 (140): . I.e. -> i.e.

Line 361: it yet learns -> it still learns

Figure 2 should be improved. A better selection of the range of value in y, can give a better insight of the results.

---

### Official Review · Reviewer_fCrV · 2022-10-21

**Overall Score:** 5
**Confidence:** 4

**Review:**

————————————————————

Summary of the work

This paper introduces a graph-time convolutional autoencoder (dubbed GTConvAE) for unsupervised learning
from multivariate time series on network graphs. The architecture exploits a space-time product graph to encode
the data structure (the graph shift operator has a learnable parametric representation), and uses graph convolutional filters
in said product graphs at both the encoder and decoder modules. Temporal downsampling is performed at the encoder
to enhance the spatiotemporal receptive field. Stability of GTConvAE with respect to relative perturbations in the spatial
graph is established. Numerical experiments show the effectiveness of the proposed autoencoder in denoising of solar
irradiance time series and anomaly detection tasks.

————————————————————

Strengths

+ The development of graph-time autoencoders for unsupervised learning from dynamic network data has received
significant attention recently. The problem addressed is timely and relevant.
+ The references and description of prior related work is comprehensive. The proposed work is adequately positioned
in context.
+ The discussion related to the upshot of the stability result in Section 3.2 is nicely done. It offers valuable insights on how the
different components of the GTConvAE architecture affect stability when the spatial graph is perturbed. Said discussion
is supported with numerical experiments in Section 4.3, by examining stability as SNR, number of nodes, and temporal
downsampling rates vary.
+ Results from the numerical experiments show that GTConvAE compares well with other graph-time autoencoders,
especially in high SNR regimes. The anomaly detection task of detecting cyber-physical attacks on a water network
is particularly interesting.

————————————————————

Weaknesses

- There are concerns on the technical novelty of the proposed approach. The GTConvAE builds on graph-time convolutional
(GTConv) filters to obtain node representations in product graphs, which have been proposed in [26]. Another interesting idea
is the adoption of the parametric product graph in (1) for added flexibility, but this appears to have been introduced in [26] as
well. Likewise, the stability analysis largely follows the setting and proof methodology adopted in several prior graph neural
network (GNN) papers. All in all, the innovations required to develop the GTConvAE model are limited.
- Often, claims used to characterize limitations of existing work or to motivate proposed architectures are arguable, loose, and
unjustified. Examples are “…fixing the spatiotemporal dependencies in the data, which may lead to wrong inductive biases”; “a symmetric
structure…, making it suitable for tasks concerning network time series”; and “By working disjointly…., these approaches fail to capture
the joint spatiotemporal dependencies present in the raw data.” Said claims should be better supported, recalibrated, or else removed.
- The problem formulation does not account for dynamic graphs, namely the graph-shift operator $\mathbf{S}$ of the spatial graph is
assumed to remain invariant over $t=1,\ldots,T$. If the graph topology changes, and say the changes are known, can these be
seamlessly accommodated in GTConvAE? While this could be beyond the scope of this paper, at least noting this limitation is
important because the Introduction states other models can accommodate dynamic topologies.
- Unlike most deep learning architectures where layer outputs (pre activation) are linear functions of the learnable parameters, here
the $\{s\_{ij}\}$ enter polynomially in the model. Since polynomial functions are notoriously hard to optimize, one wonders if this
parameterization can adversely affect training of GTConvAE.
- The problem of denoising a multivariate time series has a long history in statistics, signal processing and related areas. While it is
clear that the test case in Section 4.1 serves as a proof of concept, there are concerns as to why only graph-time auto encoders are
adopted as baselines. Moreover, why is the algorithm in [2] not used for the performance comparisons in this setting?

————————————————————

Recommendation and justification

Given the issues raised above and in particular given the limited technical innovation (in the context of a timely, but already widely
investigated problem), the paper does not merit publication in its current form. If a strong case can be made on the significance of the
contribution given the related work in [26], then I am willing to reconsider my recommendation.

————————————————————

Questions and suggestions for improvement

- Introduction, third paragraph: “The two works elaborating on this…” is unclear. What does “this” refer to here? Please rephrase.
- Can you elaborate on the novelty relative to [26]? Say when it comes to the stability result, is there any technical innovation that
is required to accommodate the autoencoder setting dealt with here? The first component of the proof establishes GTConv filters are
stable, so cannot one just reference [26] for that step?
- Based on the discussion following (4), I was wondering if the effect of downsampling cannot be accomplished by adding (length $r$)
temporal skip connections in the product graph? This could increase the spatiotemporal memory without needing to perform data
summarization.
- Section 3.1, first paragraph: it should read $\mathbf{V}\_{T}$.
- One line after (15): it should read “…response of a graph-time…” What is the “coefficient” here? Is it $C$? Please clarify.
- In Definition 2, use GTConv filter.
- What are solar networks or solar cities? Why is [2] not included as a baseline in Section 4.1?
- Section 4.3: please fix the $x-$label in Figure 2(c). Can we say that the filters used for the experiments satisfy Definitions 1 and 2?
Please provide details on how is the perturbation matrix $\mathbf{E}$ generated as part of the experimental setup.
- I would argue that a more interesting experiment is one in which the stability properties of different graph-time autoencoders are compared. How robust are the models considered to perturbations in the spatial graph?

---

### Official Review · Reviewer_PxUT · 2022-10-21

**Overall Score:** 6
**Confidence:** 3

**Review:**

### Contributions of the Paper
(i) To  improve the receptive field in a spatiotemporal manner, the authors make product graph parametric to attend to the spatiotemporal coupling for the task at hand.
(ii) From robust perspective, the authors prove that the GTConvAEs with graph integral Lipschitz filters are stable to graph perturbations.
(iii) Extensive experiments verify the efficacy of the proposed GTConvAE for the improvement over existing methods on two real-world datasets (for time-series denoising and anomaly detection tasks).

### Main Review
#### Strength:
(i) This paper is well-written and easy to follow.

(ii) The paper introduces the product graph representation of the spatiotemporal patterns which can capture information in both spatial and temporal dimensions from raw data and the intermediate higher-level representations.

(iii) The paper proposes a temporal downsampling/upsampling in the encoder/decoder to increase the spatiotemporal receptive field.

(iv) Experiments evaluation is extensive and helps understand the power of GTConvAE model.

#### Weakness:
(i) Related work and the problem definition are missing.

(ii) On page 2, the notation $Z$ (i.e., higher-level representations) are not defined in above GTconvAE Equation.

(iii) Although the complexity analysis is provided, I suggest the authors providing running time/cost.

(iv) More state-of-the-art baselines (e.g., recent spatio-temporal graph neural networks [1]) are required for the final revision.

(v) Why not evaluate the method on the commonly used benchmarks? For instance, PeMS traffic datasets and there are extensive prior work have reported performance on those datasets.

(vi) Ablation studies are missing which can help better understand the importance of each improvement in the GTconvAE model.

[1] Jiang, W. and Luo, J., "Graph neural network for traffic forecasting: A survey", Expert Systems With Applications, 2022.

---

### Decision · Program_Chairs · 2022-11-23

**Decision:**

Accept (Poster)

**Comment:**

With some concerns about novelty and related work being raised, we encourage the authors to incorporate such suggestions in the camera-ready version.